Reducing publication delay to improve the efficiency and impact of conservation science

http://orcid.org/0000-0002-8465-8410 Christie Alec P. 1 2 apc58@cam.ac.uk
White Thomas B. 1 tbw27@cam.ac.uk
http://orcid.org/0000-0002-5346-8868 Martin Philip A. 1 2
http://orcid.org/0000-0002-3984-2403 Petrovan Silviu O. 1
http://orcid.org/0000-0002-2677-1247 Bladon Andrew J. 1
Bowkett Andrew E. 3
Littlewood Nick A. 1 4
http://orcid.org/0000-0002-5671-0963 Mupepele Anne-Christine 5
http://orcid.org/0000-0003-2757-7347 Rocha Ricardo 1 6 7
http://orcid.org/0000-0002-5087-3892 Sainsbury Katherine A. 8
Smith Rebecca K. 1
Taylor Nigel G. 9
Sutherland William J. 1 2
1 Department of Zoology, University of Cambridge , Cambridge , United Kingdom
2 BioRISC, St Catherine’s College , Cambridge , UK
3 Wild Planet Trust, Paignton Zoo , Paignton, Devon , United Kingdom
4 Scotland’s Rural College (SRUC) , Craibstone Estate, Aberdeen , United Kingdom
5 Faculty of Environment and Natural Resources, University of Freiburg , Freiburg , Germany
6 Research Center in Biodiversity and Genetic Resources, Institute of Agronomy, University of Lisbon, CIBIO-InBIO , Lisbon , Portugal
7 Research Center in Biodiversity and Genetic Resources, University of Porto, CIBIO-InBIO , Vairão , Portugal
8 Faculty of Kinesiology, Sport, and Recreation, University of Alberta , Edmonton, Alberta , Canada
9 Tour du Valat, Research Institute for the Conservation of Mediterranean Wetlands , Arles , France
Stern David
Electronic publication date: 2021 Oct 12
Publication date: 2021
Volume: 9
Electronic Location ID: e12245
Received 2021 May 12; Accepted 2021 Sep 13
Copyright: © 2021 Christie et al.
Copyright year: 2021
Copyright holder: Christie et al.
License: This is an open access article distributed under the terms of the Creative Commons Attribution License, which permits unrestricted use, distribution, reproduction and adaptation in any medium and for any purpose provided that it is properly attributed. For attribution, the original author(s), title, publication source (PeerJ) and either DOI or URL of the article must be cited.
License URL: https://creativecommons.org/licenses/by/4.0/

Keywords: Publication delay, Peer-review, Conservation science, Conservation evidence, Evidence-based, Science-policy gap, Science-practice gap, Journal delay, Publication speed

Funding: Arcadia, The David and Claudia Harding Foundation and MAVA Alec P Christie was supported by the Natural Environment Research Council as part of the Cambridge Earth System Science and The David and Claudia Harding Foundation NERC DTP [NE/L002507/1] Department of Zoology, Cambridge University This work was supported by Arcadia, The David and Claudia Harding Foundation and MAVA for funding. Alec P Christie was supported by the Natural Environment Research Council as part of the Cambridge Earth System Science NERC DTP [NE/L002507/1] and The David and Claudia Harding Foundation. Thomas B White was supported by the Balfour Studentship awarded by the Department of Zoology, Cambridge University. The funders had no role in study design, data collection and analysis, decision to publish, or preparation of the manuscript.

==============================
Evidence-based decision-making is most effective with comprehensive access to scientific studies. If studies face significant publication delays or barriers, the useful information they contain may not reach decision-makers in a timely manner. This represents a potential problem for mission-oriented disciplines where access to the latest data is required to ensure effective actions are undertaken. We sought to analyse the severity of publication delay in conservation science—a field that requires urgent action to prevent the loss of biodiversity. We used the Conservation Evidence database to assess the length of publication delay (time from finishing data collection to publication) in the literature that tests the effectiveness of conservation interventions. From 7,447 peer-reviewed and non-peer-reviewed studies of conservation interventions published over eleven decades, we find that the raw mean publication delay was 3.2 years (±2SD = 0.1) and varied by conservation subject. A significantly shorter delay was observed for studies focused on Bee Conservation, Sustainable Aquaculture, Management of Captive Animals, Amphibian Conservation, and Control of Freshwater Invasive Species (Estimated Marginal Mean range from 1.4–1.9 years). Publication delay was significantly shorter for the non-peer-reviewed literature (Estimated Marginal Mean delay of 1.9 years ± 0.2) compared to the peer-reviewed literature (i.e., scientific journals; Estimated Marginal Mean delay of 3.0 years ± 0.1). We found publication delay has significantly increased over time (an increase of ~1.2 years from 1912 (1.4 years ± 0.2) to 2020 (2.6 years ± 0.1)), but this change was much weaker and non-significant post-2000s; we found no evidence for any decline. There was also no evidence that studies on more threatened species were subject to a shorter delay—indeed, the contrary was true for mammals, and to a lesser extent for birds. We suggest a range of possible ways in which scientists, funders, publishers, and practitioners can work together to reduce delays at each stage of the publication process.

Introduction

Across many mission-oriented disciplines that respond to urgent societal issues, evidence-based decision-making is critical to improving the effectiveness and efficiency of practice. This requires comprehensive access to scientific studies providing data useful for judging the likely effectiveness of actions to achieve desired outcomes. New scientific studies not only add to the relevant corpus of information that can guide decisions, but are likely to be particularly influential due to continually evolving technologies, methodologies, and skills, as well as the focus on issues of current concern. However, if new studies are not made available, or delayed in being so, relevant information for decision making (i.e., evidence; Salafsky et al., 2019) may not reach decision-makers quickly enough to prevent harmful or ineffective decisions being made.

Biodiversity conservation is motivated by a need for rapid, transformative societal change to tackle biodiversity loss (Mace et al., 2018; Leclère et al., 2020). To do this, we must dramatically improve and upscale conservation efforts by ensuring that they are as effective and efficient as possible, and based on the best relevant evidence (Sutherland et al., 2004; Pullin & Knight, 2009). However, despite progress in the assessment of the effectiveness of conservation interventions (e.g., Sutherland et al., 2019), the evidence base remains patchy (Christie et al., 2020, 2021), as many commonly used interventions remain understudied and evidence for some threatened taxa or habitats, and some geographic regions, remains non-existent or minimal for relevant conservation actions (e.g., Taylor et al., 2019; Junker et al., 2020). To bolster the evidence base for conservation, we must rapidly test and disseminate findings on the effectiveness of different conservation actions.

The urgency of biodiversity decline, and the fast-paced nature of many conservation issues, mean that large delays in publishing evidence could have detrimental impacts. Without prompt access to information on conservation status, threats and responses, we risk misspending funds on activities that are inefficient, ineffective, suboptimal or even harmful for biodiversity. For example, global wind energy infrastructure has expanded rapidly from a capacity of 489,000 MW in 2016 to 651,000 MW in 2019 (https://library.wwindea.org/global-statistics/), and delays to research papers testing cost-effective interventions to minimise bird and bat collisions could miss key opportunities to mitigate the impacts of this expansion. In addition, there are many examples of rapid species’ decline where targeted research is needed to stop and reverse threatening processes (Williams, Balmford & Wilcove, 2020). For instance, between 1982 and 1984, Northern White Rhinos declined from several hundred individuals, to just over ten, with published studies on rhino biology and management only increasing several years after the decline (Linklater, 2003).

The problem of publication delay appears to be particularly acute in conservation. Kareiva et al. (2002) found that the mean time between the date of submission to the journal in which an article was eventually published (i.e., the destination journal) and the date of publication was 572 days in conservation science, far higher than for studies in genetics and evolutionary biology which had an average delay of 249 days. In 2009, a similar assessment looked at the same conservation journals, and found a destination journal delay of 402 days, which was still higher than biological fields such as taxonomy (335 days), behaviour (379 days), and evolution (181 days), but had significantly declined over the previous seven years (O’Donnell, Supp & Cobbold, 2010). The same study investigated the delay between the dates of final data collection and article submission, and found a median delay of 696 days—as before, this was higher than for other biological fields (taxonomy = 605 days; behaviour = 508 days; evolution = 189 days; O’Donnell, Supp & Cobbold, 2010). If this same trend holds for studies that test the effectiveness of conservation interventions, we would expect that three years would pass before such studies can help to inform the conservation community on the effectiveness of an intervention (without accounting for issues of freely accessing the published paper).

To examine the extent of this problem, specifically in the literature that tests conservation interventions, we investigate: 1. The length of publication delay in studies that test the effectiveness of conservation interventions;

2. How publication delay: (a) differs between different conservation subjects;

(b) has changed over time; and

(c) differs between different publication sources (i.e., peer-reviewed and non-peer-reviewed literature);

3. For species-focussed studies, how publication delay changes with the IUCN Red List (IUCN, 2020) status of the species on which interventions are tested; and

4. The typical length of manuscript processing at destination journals.

We define publication delay as the time taken from finishing data collection for a study to when the study is published (either in a peer-reviewed scientific journal or in the non-peer-reviewed literature). We discuss the factors that could be driving publication delay, and provide suggestions on how the scientific community can work together to minimise them.

Methods

Calculating publication delay

Using the Conservation Evidence database (Sutherland et al., 2019), we examined the difference between the year that data collection ended for a study and the year that the study was published (either in a peer-reviewed scientific journal or in the non-peer-reviewed literature). The Conservation Evidence database contained studies documenting 2,399 conservation interventions (as of December 2020; e.g., sowing strips of wildflower seeds on farmland to benefit birds) across multiple ‘synopses’. Synopses are used in the Conservation Evidence database to categorise studies into useful subject areas such as by species group, habitat, or related interventions (e.g., ‘Bird Conservation’ or ‘Management of Captive Animals’).

To construct the Conservation Evidence database, publications were retrieved from the literature using a standardised protocol of manual searching through entire journals, and non-journal literature sources, for quantitative assessments of the effectiveness of a conservation intervention (‘subject-wide evidence synthesis’; see Sutherland et al., 2019 for details). The aim was to cover all the main general and specific journals; as of July 2021 this has involved reading 1,629,688 paper titles from 657 journals. We focused on the number of unique studies of an intervention within each Conservation Evidence synopsis. For example, if a publication reports studies of two different interventions (e.g., supplementary feeding and provision of artificial nests), then these studies are counted separately. Using this classification of conceptually distinct studies, we were able to extract information on when 7,477 studies were published and when their data collection ended. Only approximately 1% of studies did not report dates (68 out of 8,209 in the entire database) and so were excluded from the analyses. The remaining 664 studies were reviews or meta-analyses, which were excluded as we were only interested in the publication delay of primary literature.

We extracted temporal information from the Conservation Evidence database (publication year) and a summary of each study that included information on the years during which the study was conducted. We defined the end year of a study as the year within which data collection ended. End years were extracted from Conservation Evidence summaries using regular expressions and text mining of the website (www.conservationevidence.com) with the XML package (Lang, 2020a) and RCurl package (Lang, 2020b) in R statistical software version 3.5.1 (R Core Team, 2020). This extraction was necessary because this information is not currently stored in the database. We checked the accuracy of text mining by reviewing data for a random sample of 382 studies (approximately 5% of the total number of studies analysed) and found that 94.4% had the correct study end year. Although there were a small number of errors, these were mostly caused by assigning the publication year as the study end year, and therefore would yield an underestimate of publication delay. In addition, automating the extraction of dates from study summaries offered the most feasible and reproducible way to analyse the entire database, and avoided human error and unconscious bias that would affect manual extraction of dates from a large database (Christie et al., 2020, 2021).

To determine publication delay, we subtracted the end year of each study from its publication year. For studies conducted and published within the same year, their length of publication delay was therefore zero years. The coarse temporal resolution of years would have caused us to overestimate publication delay for studies with a delay of a few months that run between calendar years (e.g., December 2000 to March 2001), but underestimate the delay for studies published in months that do not span calendar years (e.g., January 2001 to December 2001). Across many studies these effects should generally cancel out—although rounding down of studies completed within one calendar year makes our overall estimation of publication delay conservative. The publication year we used typically described when studies were published in an issue, which may mean that we overestimated publication delay for a small number of studies. However, it has been found in other disciplines (urology and nephrology) that early view articles in 2014 were, on average, published only 95 days earlier than the final published article (Echeverría, Stuart & Cordón-García, 2017)—a similar delay of approximately 3 months, applicable to only a subset of studies, is unlikely to substantially alter our results.

Testing for variation between conservation subjects, over time, and between publication sources

We used a Generalised Linear Model (GLM) to quantify and statistically test how publication delay varied: (i) between different synopses (Amphibian Conservation, Bat Conservation, Bee Conservation, Bird Conservation, Control of Freshwater Invasive Species, Farmland Conservation, Forest Conservation, Management of Captive Animals, Mediterranean Farmland, Natural Pest Control, Peatland Conservation, Primate Conservation, Shrubland and Heathland Conservation, Soil Fertility, Subtidal Benthic Invertebrate Conservation, Sustainable Aquaculture, Terrestrial Mammal Conservation); (ii) over time (by publication year); and (iii) between the peer-reviewed and non-peer-reviewed literature. See Table S2 for the number of studies in each synopsis.

Therefore, we used three explanatory variables (synopsis, peer review, and publication year) to predict the response variable of publication delay. As publication delay was a count variable (non-negative integers), we first tested a Poisson GLM, but this had a dispersion parameter value of 2.85 and so we used a quasi–Poisson GLM instead whereby the standard errors are corrected for overdispersion (using variance = mu * theta, where mu is the mean of the dependent variable distribution, and theta is the dispersion parameter of the quasi–Poisson model). The synopsis ‘Bee Conservation’ and non-peer reviewed category were set to the zero value of the beta slope and used as reference categories as these had the lowest mean publication delay values based on preliminary exploration of the data. We selected the best model structure using quasi–Akaike’s Information Criterion (qAIC) (see results; Table S3).

We used the R package emmeans (Lenth, 2021) to calculate the Estimated Marginal Means (EMMs—or least squares means using the Tukey adjustment) and asymptotic 95% Confidence Intervals (using the Sidak adjustment) for different synopses and types of publication sources, as well as to test for significant differences between these categories based on the quasi–Poisson GLM. We used the EMMs (which average across the values of other explanatory variables or fix these variables at a constant value) and their associated 95% Confidence Intervals for different categories to produce most data visualisations and summary statistics, except for calculating publication delay for all studies combined for which we used a simple raw mean.

Since studies that are published more recently have a greater potential to suffer from a longer delay on a purely mathematical basis—i.e., a study published in 2020 has theoretically had 10 more years over which it could have remained unpublished than a study published in 2010—we conducted a sensitivity analysis to check for this potential bias. We restricted the dataset we analysed in our original GLM to different time periods: 1980–2020, 1990–2020, 2000–2020, and 2010–2020 and repeated GLMs (Table S4).

To test for differences between studies published in the peer-reviewed and non-peer-reviewed literature, we classified the publication type of studies in the Conservation Evidence database using two categories: (i) peer-reviewed sources; and (ii) non-peer-reviewed sources. This differentiation of publication sources by peer-review allowed us to approximate the degree of publication delay that is due to the peer-review and editorial process in journals—assuming that similar studies are published in both types of literature sources. We used the Scimago (2020) dataset to detect recognised peer-reviewed journals and then manually searched the websites and/or texts of any unrecognised publication sources to check whether they were peer-reviewed or not; if there was no explicit evidence of peer-review, we allocated studies to the ‘non-peer-reviewed’ literature category. Whilst some books, reports, and articles published outside of scientific journals may undergo a form of internal peer review (potentially within organisations), we focused on defining peer-review as that occurring within the context of a scientific journal or magazine where there was explicit evidence of a peer-review process. Therefore, a small number of publication sources that undertake some form of non-journal peer-review may have been allocated to the non-peer-reviewed category. However, the number of studies affected is likely to be small given that such peer-review is typically uncommon in non-journal sources based on our experience. For names of all publication sources in each of the categories, see Table S1.

Testing for differences in publication delay between IUCN red list categories

In a separate analysis, we tested for significant differences in publication delay between studies testing interventions on species with different IUCN Red List statuses. We first extracted data from the Conservation Evidence database on the species studied within five taxa-specific synopses (n = 3,941 studies in: Amphibian Conservation, Bird Conservation, Terrestrial Mammal Conservation, Primate Conservation, and Bat Conservation synopses), and the threat status of each species from the IUCN Red List (IUCN, 2020). We limited the analysis to these synopses as these taxa had been comprehensively assessed in the IUCN Red List. We ran separate quasi–Poisson GLMs on this reduced dataset, and then separately for each of the three broad taxa (amphibians = Amphibian Conservation; birds = Bird Conservation; mammals = Terrestrial Mammal Conservation, Primate Conservation, and Bat Conservation synopses). We included explanatory variables of publication source, publication year, and IUCN Red List Status for our initial models (plus the term synopsis for mammals to maintain consistency with our main model as this taxon has studies split across three different synopses). We selected the best model structure for each subset of data using qAIC values (see results for model structures selected; Table S5)—if two models were within <2 units of qAIC, we selected the simplest model (according to the principle of parsimony).

For these taxonomic GLMs, we only considered the most threatened IUCN Red List category (out of Least Concern, Near Threatened, Vulnerable, Endangered, Critically Endangered) of all species for each published study. For example, if a study targeted multiple species, such as two that were listed as Least Concern and one listed as Endangered, we considered that as a study on an Endangered species. To determine whether picking the most threatened species in each study affected our results, we conducted a sensitivity analysis through rerunning analyses and selecting a random species from each study (see results; Tables S6 and S7).

There were insufficient studies on species with IUCN Red List statuses of Data Deficient (four studies) or Extinct in the Wild (eight studies) and so we did not include these categories in our taxonomic analyses. Five mammal species could not be matched to the IUCN Red List (e.g., domesticated or hybrid species) and so were not represented in the analysis (this only affected 18 studies or 0.46% of studies in the taxonomic analysis). We were unable to obtain the IUCN Red List status of species at the time when each relevant study was conducted and therefore had to use the current status of species in the latest IUCN (2020) Red List update. Whilst this may mean that, for some studies, certain species may have changed in their Red List status in the intervening years, the current threat category is an indication of the need for previous studies on responses that could have helped prevent this decline assuming that, for many species, threatening processes have been present over long time-periods.

Investigating journal processing delays

To investigate delays taking place during manuscript processing by journals (i.e., from submission to a journal to online publication of the study), we extracted data on the number of days from receipt of studies to acceptance and the number of days from acceptance to online publication for any study published from 2015–2021 and indexed in PubMed, using code produced by Himmelstein & Powell (2016); see also Himmelstein (2016). We restricted the dates to between 2015–2021 to capture recent data on journal processing delays. We then filtered these data to only include journals present in our Conservation Evidence dataset, as well as those that contained the terms ‘Wildlife’, ‘Environment’, ‘Ecology’, ‘Biology’, ‘Conservation’, OR ‘Zoology’—we also manually edited the list of retrieved journals by removing any journals that we believed were obviously not relevant venues for conservation science studies (e.g., Free Radical Biology and Medicine). Journals and studies for which data can be extracted is dependent on their deposition in PubMed—the number of studies at different journals for which delays could be calculated is therefore variable and sometimes small. This also only gives a general view of delays taking place at journals for any given study, rather than those specifically focusing on conservation interventions—however, we believe these data can still provide some general guidance on the publishing speed of journals relevant to researchers wishing to publish studies that test conservation interventions (Table S14).

R code to perform all analyses is available here: https://doi.org/10.5281/zenodo.4621310.

Results

Extent and variation of publication delay (objective 1 & 2)

The simple raw mean publication delay of studies of conservation interventions across all Conservation Evidence synopses was 3.24 years (95% Confidence Intervals = [3.17, 3.31]; Fig. 1)–note that means presented after this point are Estimated Marginal Means (EMMs), which are adjusted for other confounding variables in our models.

Figure 1 Distribution of studies of conservation interventions according to the length of publication delay (in years) for different Conservation Evidence synopses (each covering a distinct conservation subject—e.g., ‘Bird Conservation’).

Solid red vertical lines indicate the mean length of publication delay for each plot and dashed red lines represent 95% Confidence Intervals. For each synopsis (B), summary statistics were obtained using Estimated Marginal Means (averaging over other explanatory variables) based on a quasi–Poisson Generalised Linear Model (GLM), whilst for all synopses combined (A) a simple raw mean was obtained from an intercept-only quasi–Poisson GLM.

A full model containing all three explanatory variables (publication date, peer-review category, and synopsis) was selected because it had the lowest qAIC score (Tables S3; S8). When accounting for these variables using EMMs, publication delay varied significantly between several synopses (p < 0.001 to p = 0.049; Fig. 1). Most notably, Bee Conservation, Sustainable Aquaculture, Management of Captive Animals, Amphibian Conservation, and Control of Freshwater Invasive Species had a significantly shorter mean delay (1.41–1.85 years; Fig.1; see also Table S2 for delays by synopsis) than most other synopses (2.20–3.33 years; range of synopses’ significant p-values: p < 0.001 to p = 0.044; see also Fig. S1 and Tables S8, S9 for detailed statistics and comparisons).

There was a statistically significant increase in publication delay from 1912 to 2020 of 1.21 years (from EMMs of 1.40 to 2.61 years of delay; t = 4.46; p < 0.001; Fig. 2; Table S8). Sensitivity analyses demonstrated that there was a statistically significant increase in publication delay since 1980 (estimate = +0.51 years from 1980–2020; t = 3.59; p < 0.001) and 1990 (estimate = +0.35 years from 1990–2020; t = 2.6; p = 0.009), but not for the period since 2000 (estimate = −0.08 years from 2000–2020; t = 0.6; p = 0.556), or 2010 (estimate = +0.21 years from 2010–2020; t = 0.7; p = 0.467; Table S4).

Figure 2 Changes in publication delay relative to the year in which studies of interventions were published.

The shade of hexagons is relative to the number of data points (studies) at that position on the graph. The red solid and dotted lines represent Estimated Marginal Means (averaging over other explanatory variables) and associated 95% confidence intervals based on a quasi–Poisson Generalised Linear Model (GLM) for publication delay (see Table S8 for full model result). Only data for a publication delay of 20 years or less is presented to improve visualisation, but all data were used in the GLM (see full data figure Fig. S2). We conducted sensitivity analyses to check whether the trend changed in more recent decades (see Table S4).

Publication delay also differed significantly by publication source (t = 6.8, p < 0.001; Fig. 3); studies from non-peer-reviewed literature (mean delay of 1.90; 95% Confidence Intervals = [1.64, 2.20]) had a significantly shorter delay than studies published in peer-reviewed literature (mean delay of 2.95; 95% Confidence Intervals = [2.84, 3.07]; z = −6.8; p < 0.001; Tables S8; S10).

Figure 3 Publication delay in years for all studies of interventions published in the peer-reviewed and non-peer-reviewed literature.

Solid red vertical lines indicate the mean length of publication delay for each type of publication source and dashed red lines represent 95% Confidence Intervals. For each category, summary statistics were obtained using Estimated Marginal Means (averaging over other explanatory variables) based on a quasi–Poisson Generalised Linear Model (GLM). For publication sources classified under each category, see Table S1.

IUCN red list status (objective 3)

In a separate taxonomic analysis, we considered whether publication delay varied by the IUCN Red List status of species on which interventions were tested (Figs. 4 & 5). When pooling studies testing interventions on amphibians, birds, and mammals (Fig. S3), there were significant differences in publication delay between IUCN Red List status categories (t = 3.1 to 11.9; p < 0.001 to p = 0.002; Fig. 4A; Table S11), whereby more highly threatened categories appeared to generally suffer from a longer publication delay. Studies on Least Concern species had a significantly shorter delay than Critically Endangered species (z = −3.1; p = 0.015) and Endangered species (z = −11.9; p < 0.001), whilst studies on Near Threatened species had a significantly shorter delay than Endangered (z = −9.4, p < 0.001) and Critically Endangered species (z = −3.1, p = 0.015; Fig. 4A; Table S12). Studies on Vulnerable species also had a significantly shorter delay than Endangered (z = −10.2, p < 0.001), and Critically Endangered species (z = −3.6, p = 0.003), whilst Endangered species had a significantly shorter delay than Critically Endangered species (z = −3.8, p = 0.001; Fig. 4A; Table S12).

Figure 4 Z-ratios and p-values from post-hoc statistical tests of differences in publication delay between IUCN categories for all amphibians, birds, and mammals (A) amphibians (B) birds (C) and mammals (D).

Darker red or blue coloured cells indicate greater Z-ratios (blue = positive difference, red = negative difference) and larger diamonds indicate smaller p-values, whilst red coloured diamonds indicate p-values of p < 0.05 (black diamonds indicate p ≥ 0.05). For example, for mammals only (D), studies on Least Concern species have a significantly shorter mean delay than studies on Endangered species (top row, second column is dark red with red diamond), but did not have a significantly shorter mean delay compared to studies on Vulnerable species (top row, third column is grey with black diamond). Post-hoc tests of differences between synopses were conducted using Estimated Marginal Means with Tukey adjustment in the R package emmeans (Lenth, 2021).

Figure 5 Publication delay of studies of conservation interventions (in years) grouped by the IUCN Red List Category of the species that were studied.

Data presented for amphibians (Amphibia from the Amphibian Conservation synopsis), birds (Aves from the Bird Conservation synopsis), and mammals (Mammalia from the Bat Conservation, Primate Conservation, and Terrestrial Mammal Conservation synopses). IUCN threatened categories include Vulnerable, Endangered, and Critically Endangered, whilst non-threatened categories include Least Concern and Near Threatened (following IUCN Red List; 2020). We did not include the few studies on Data Deficient and Extinct in the Wild species (see Methods). Vertical solid lines show mean publication delay and dashed lines show 95% Confidence Intervals. For each taxonomic group (amphibians, birds, and mammals), summary estimates were obtained using Estimated Marginal Means (averaging over other explanatory variables) based on quasi–Poisson Generalised Linear Models (GLMs).

Considering amphibian IUCN Red List status separately, the best and most simple model structure chosen by qAIC only included the literature source and publication date as explanatory variables (Table S5)—tests of pairwise differences between EMMs using the full model for amphibians (i.e., including IUCN Red List category) found no significant differences in the mean delay of studies for different IUCN Red List categories (z = |0.001| to |1.1|; p = 0.829 to 1.000; Figs.4B & 5; Table S12). For birds, the best and most simple model structure chosen by qAIC only included IUCN Red List category as an explanatory variable (Table S5), which showed significant differences between several IUCN Red List categories (z = |3.4| to |7.5|; p < 0.001 to p = 0.005; Figs.4C & 5; Table S12). For mammals, a full model (including synopsis as mammal species are split into several synopses—Bat, Primate, and Terrestrial Mammal Conservation) was selected using qAIC and there were several significant differences between IUCN Red List categories (z = |3.4| to |9.8|; p < 0.001 to p = 0.006; Fig. 4D; Table S12). Therefore, differences between IUCN categories shown in the full model (pooling studies on amphibians, birds, and mammals) appeared to be driven by studies on mammals and birds.

For mammals, studies on Least Concern species, Near Threatened, and Vulnerable species had a significantly shorter delay than Endangered and Critically Endangered species (z = −3.4 to −9.8; p = 0.006 to p < 0.001), but there were no other significant differences between IUCN Red List categories (z = |0.3| to |1.5|; p = 0.529 to 0.999; Figs. 4D & 5; Table S12). For birds, patterns were inconsistent in their direction: studies on Least Concern species had a significantly shorter delay than Endangered species (z = −6.6; p < 0.001; Figs. 4C & 5; Table S12)—although not compared to Near Threatened (z = 0.6; p = 0.976) or Critically Endangered species (z = 1.8; p = 0.358). Studies on Vulnerable species had a significantly shorter delay compared to Least Concern species and Endangered species (z = −3.4 to −7.5, p < 0.001 to p = 0.005), but not compared to Near Threatened and Critically Endangered species (z = 0.0 to 2.7; p = 0.050 to 1.000), whilst studies on Critically Endangered species had a significantly shorter delay than for Endangered species (z = −4.9; p < 0.0001; Figs. 4C & 5; Table S12). Studies on Near Threatened species also had a significantly shorter delay than Endangered species (z = −6.3; p < 0.001; Figs. 4C & 5; Table S12).

We conducted a sensitivity analysis to determine whether selecting the species with the most threatened IUCN status from each study may have affected our results by rerunning the taxonomic analyses and instead selecting a random species from each study, but the results changed negligibly (Tables S6, S7 & S13).

Journal processing times (objective 4)

For journals indexed in PubMed that were included in our dataset, the time from submission to acceptance was on average 150 days (95% Confidence intervals = [139, 161]), and acceptance to publication was on average 55 days (95% CIs = [46, 67]). When adding the mean delays from submission to acceptance and acceptance to publication together for each journal, the mean destination journal delay (time from submission to publication; see ‘Breaking it down: considering different components of publication delays’) was 186 days (95% CIs = [169, 205]).

Figure 6 Typical publication timeline to define publication delay for studies submitted to journals (and other non-journal sources) and categorise different types of delay.

The first and second journals/sources are considered intermediary prior to being submitted to the destination journal/source where the study is accepted and published. Write-up delay and resubmission delay are often combined and known collectively as ‘submission delay’ in studies investigating publication delay. Studies published in the non-peer-reviewed literature would mainly suffer from write-up delay, as such studies typically progress straight to acceptance and publication (sometimes after some administrative and editorial processes).

Expanding the journals included in these estimates to those with a name containing the terms ‘Wildlife’, ‘Environment’, ‘Ecology’, ‘Biology’, ‘Conservation’, or ‘Zoology’ (also filtered manually for journals that have relevance to conservation science—see Methods), delays were relatively similar, albeit slightly shorter (time from submission to acceptance = 130 days (95% CIs = [125, 136])), time from acceptance to publication = 47 days (95% CIs = [43, 52]), and overall mean destination journal delay = 166 days (95% CIs = [159, 175]; see Table S14 for delays by journal).

Discussion

The nature of publication delays in conservation science

Our results show that conservation decision-makers must typically wait over three years after data collection has finished for the latest evidence on the effectiveness of conservation interventions to be published. Studies published in the non-peer-reviewed literature had a shorter delay than those published in the peer-reviewed literature. Although we found publication delay has increased since 1912, sensitivity analyses suggested that change post-2000s was weak. Finally, we found little evidence that studies focusing on species listed as more threatened by the IUCN Red List suffered from shorter delays, with results to the contrary for mammals and birds.

Our results concur with previous analyses of publication delay in the wider conservation literature (average delay of three years; O’Donnell, Supp & Cobbold, 2010) which found longer delays than the other disciplines considered (e.g., one year for evolution, 2.6 years for taxonomy and three years for behaviour; O’Donnell, Supp & Cobbold, 2010). Whilst variation in delays may be expected depending on the nature of scientific work, large and variable delays have been found in other scientific areas, including mission-driven disciplines that would benefit from timely dissemination of results. For example, studies have shown a mean destination journal delay (time from submission to publication) of 9.5 months in biomedicine, nine months for chemistry and engineering, 14 months for social sciences and 17 months for business and economics (Björk & Solomon, 2013). Similarly, studies have found that, on average, 10 months passed between the release of a press statement and the publication of trial results in oncology (Qunaj et al., 2018), and that only 53% of vaccine trials were published within three years after trial completion (Manzoli et al., 2014). These cross-disciplinary findings are particularly problematic for evidence syntheses, which in turn take time to produce and disseminate. Any delays to the publication of primary research studies will feed through into delays in updating the recommendations of evidence syntheses to guide evidence-based practice.

In conservation, a great deal can happen in three years. A species’ population may drastically decline and new threats may emerge, forcing conservationists to take rapid action to avert biodiversity and habitat loss. Without faster access to up-to-date evidence on the effectiveness of conservation actions, there is a risk that practitioners pursue ineffective practices and mis-allocate conservation resources at a time when we cannot afford to do so. Our findings are particularly concerning given that our relatively coarse estimate of publication delay (using differences between years) is likely to be conservative, since studies completed and published within a calendar year were rounded to zero.

We did identify, however, that studies on Bee Conservation, Sustainable Aquaculture, Management of Captive Animals, Amphibian Conservation, and Control of Freshwater Invasive Species had a significantly shorter mean delay (Estimated Marginal Means: 1.41–1.85 years; Fig. 1) compared to other conservation subjects. One possible explanation may be that authors in these areas target a relatively narrow pool of specialist journals for publication, with relatively high chances of acceptance during the first submission. However, there are also likely to be other reasons, such as faster publication times for particular journals in different subjects, or that certain researchers in different fields are faster at writing-up or revising their manuscripts. Ultimately, a combination of reduced time to submission, fewer resubmissions, and faster journal publication processes probably led to the shorter delays observed in certain conservation subjects.

Breaking it down: considering different components of publication delays

To better understand and minimise publication delay it is useful to distinguish the potential sources of delay, namely: (1) write-up delay (the time taken from finishing data collection to submitting a study to the first journal); (2) resubmission delay (the time taken from submitting to the first journal to submitting to the destination journal); and (3) destination journal delay (time taken from submitting to the destination journal to the publication of the study (note that these may also apply to non-journal sources; see Fig. 6)).

Delays in the publishing system (including resubmission and destination journal delay; Fig. 6) have received much attention, with calls to speed up the review and publishing of papers in conservation (e.g., Meffe, 2001; Whitten, Holmes & MacKinnon, 2001; Kareiva et al., 2002). Many journals have now worked to reduce processing times, and increase the efficiency of peer-review, by reducing unnecessary requirements for authors and making final manuscripts available early online (e.g., ‘early view’ prior to being published in an issue). Previous studies found that destination journal delay reduced in conservation from 572 days to 402 days between 2002 and 2007 (Kareiva et al., 2002; O’Donnell, Supp & Cobbold, 2010). Despite this, as the peer-review system works on a voluntary basis, delays may still be caused by the inability to find suitable reviewers for specific topics, or reviewers having insufficient time for punctual review. In the next section and later in the paper, we outline some possible ways in which the peer-review system could be changed (note these are not endorsements, but suggestions)—for example, by incentivising peer-review, allowing authors to set the timetable for peer-review, and enhancing and broadening the pool of reviewers.

Table 1 Possible solutions to reduce publication delay for studies of conservation actions.

Possible solutions	Conservation actor(s)	Delay components	
Conservation scientists and practitioners	Journals and publishers	Funders and organisations	Write-up delay	Re-submission delay	Destination journal delay	
Collaboration between scientists and practitioners to design experiments and publish results. Research organisations sometimes have time and money to write up and publish results.	✓			✓			
Buddy schemes match up individuals with others with time and knowledge suitable for analysing, reviewing and writing up results.	✓			✓	✓		
Less strict formatting and structure requirements for initial submission.		✓		✓	✓		
Journals produce article templates (e.g., as for Conservation Evidence Journal and Oryx).		✓		✓	✓		
Journals produce article types better suited for the rapid publication of tests of interventions (e.g., Research Notes, ‘Evidence’ articles in Conservation Science and Practice, Conservation Evidence Journal).		✓		✓	✓		
Provide assistance to individuals not writing in their first language to help with addressing reviewer’s comments, editing and write-up (e.g., copy-editing assistance provided by Oryx).		✓		✓	✓		
Offer pre-registration or publication of registered reports to help speed up analyses and write-up when authors finish agreed methods of data collection (Parker, Fraser & Nakagawa, 2019).		✓		✓			
Include funding, and time for writing up, in budgets.	✓		✓	✓	✓		
Include and accept published papers as project outcomes instead of reports.			✓	✓	✓		
Build a culture that values the creation of evidence-base and timely dissemination of results through training in evidence-based methods, and scientific write up.	✓	✓	✓	✓	✓		
Provide access to clear, standardised guidelines for writing up scientific articles. e.g., Oryx Writing for Conservation Guide (Oryx, 2019).		✓	✓	✓	✓		
Authors “calibrate” submissions to journals best suited to their work (Vosshall, 2012) to avoid lengthy rejections and resubmissions-e.g., using Journal/Author Name Estimator (JANE: https://jane.biosemantics.org/).	✓				✓		
Authors publish pre-prints online (e.g., BioRxiv, EcoEvoRxiv, SocRxiv) when the work has been submitted to a journal. However, caution should be taken if disseminating results due to the lack of peer-review, which is often vital for ensuring the quality and reliability of published findings.	✓				✓		
Authors pre-register study designs and/or analyses before undertaking data collection where possible (or submit a registered report). This could help plan for subsequent analysis and write-up, reduce the likelihood of rejection and the need for lengthy resubmissions and revisions due to poor quality study design or analyses (Parker, Fraser & Nakagawa, 2019).	✓			✓	✓		
Adoption of one submission models (e.g., ‘Peer Community In’) that provide peer-review that multiple publishers can access, and link papers with interested journals who use other’s reviews to guide their decisions. One such initiative, Peerage of Science, did not receive sufficient interest or support from journals and authors—it has recently been abandoned according to its founders (J Kotiaho 2021, personal communication, 4th May).	✓				✓		
Reduce unnecessary effort required for initial submissions—e.g., universal formatting and styles for submissions, word counts, flexibility in section layouts, pre-submission enquiries etc.		✓		✓	✓	✓	
Reduce time spent in unnecessary rounds of review through quick rejections, and decisive editorial decisions.		✓			✓	✓	
Incentivise peer review e.g., payments or free subscription, awards for fast, high-quality reviews (Nguyen et al., 2015), or giving reviewer’s their own DOI (if reviews are transparent; Stern & O’Shea, 2019).		✓			✓	✓	
Allow authors to submit to multiple journals simultaneously to increase competition between journals to reduce publication delay (Torgerson et al., 2005).		✓			✓	✓	
Consider providing strict deadlines to peer reviewers to promote timely returns of reviews. Encourage individuals to provide recommendations for other reviewers if they are not able to meet deadlines.		✓			✓	✓	
Consider consulting a wider pool of reviewers, and training graduate students in peer review (Nguyen et al., 2015).		✓			✓	✓	
Offer pre-registration or publication of registered reports to help the likelihood of rejection and the need for lengthy resubmissions and revisions due to poor quality study design or analyses (Parker, Fraser & Nakagawa, 2019).		✓			✓	✓	
In time-critical cases, use preliminary peer-review before submission where journals pre-identify referees in advance (e.g., fast-tracked papers in Biological Conservation; Biological Conservation, 2021) and/or send drafts to reviewers pre-review to allow reviewers to prepare comments (Sutherland & Lythgoe, 2020).		✓			✓	✓	
Once accepted, publish quickly (e.g., early view, online publishing) to reduce the time spent in publication limbo.		✓			✓	✓	
Embrace new initiatives of transparent peer-review to: share reviews between potential publishers, identify papers of interest and quickly publish the already reviewed articles.		✓			✓	✓	
Move to a peer-review system that is “publish first, curate second” through strengthening and increasing the use of preprint servers, allowing open, transparent peer-review, and the development of curation journals to select those articles of interest for specific audiences (Stern & O’Shea, 2019). This has been realised during the COVID-19 pandemic with the creation of RR:C19 a journal that rapidly and transparently reviews and curates pre-prints (Dhar & Brand, 2020).		✓			✓	✓	
Consider the use of accept/reject submission models where articles are reviewed and either accepted or rejected at the outset. Such models are used in other disciplines (e.g., Economic inquiry, Journal of Labour Research).		✓			✓	✓	
Promote the use of platforms and journals that have taken steps to reduce publication delay in the publication and peer-review process.			✓			✓	
Select platforms and journals that have taken steps to reduce publication delay in the publication and peer-review process. Many journals publish time from submission to acceptance on their websites, and resources exist to compare publication times across journals—see Table S14 and a blog post by Himmelstein (2016) from which this was derived.	✓					✓	
Consider the use of submission models (such as ‘Peer Community In’ or ‘Octopus’) that provide transparent peer-review and recommendation of pre-prints or initial submissions, but without the requirement for, although compatible with, journal publication.	✓					✓	

A more systemic problem, however, is likely to be the combination of write-up delay and resubmission delay, which are collectively known as ‘submission delay’ (Fig. 6). O’Donnell, Supp & Cobbold (2010) found a median submission delay of 696 days (1.91 years), higher than the 402 days (1.1 years) of destination journal delay they observed. We found that journals that have published tests of conservation interventions (based on our dataset) had an average destination journal delay (from submission to publication) of 186 days (0.51 years) between 2015–2021. If these delays generally hold true for the studies testing conservation interventions that are published in peer-reviewed journals in our dataset (which based on our results have a mean total publication delay of 2.95 years), a crude estimate would suggest that destination journal delay makes up only 17% of total publication delay, with submission delay accounting for the remaining 83% (i.e., write-up and resubmission delay; Fig. 6). However, this must be treated with caution as we are assuming destination journal delays have not changed greatly over time and do not vary by the type of studies submitted to these journals (i.e., intervention vs non-intervention studies). Reported destination journal delays may also be underestimates as there is the potential for journals to give ‘reject and resubmit’ decisions instead of requesting a major revision, which effectively resets the date of submission to the date of resubmission.

Nevertheless, if submission delays do make up the majority of overall publication delays, it is important to understand why these occur. Reasons for submission delay can be split into: (i) a lack of time, resources, or incentives to write up manuscripts in the conservation community; and (ii) the time-consuming nature of the preparation, formatting, referencing, peer review, and resubmission of manuscripts. Anecdotally and from our own experiences, we also suggest that issues surrounding submission delay extends to the loss of many potential papers that never progress past the stage of write-up, let alone submission or acceptance (i.e., leading to ‘infinite publication delay’). Furthermore, the longer the submission delay, the more time and effort that is required to bring manuscripts up-to-date, and the greater the likelihood of any study remaining unpublished.

Authors publishing studies of conservation interventions tend to be either conservation scientists in academia and conservation organisations or conservation practitioners who have tested interventions as part of their projects. When discussing the need for timely scientific contributions, Meffe (2001) suggested that “those with talents in and value to this field are seriously overcommitted”. Academics have to split their time between teaching, grant-writing, research projects, tutoring, and other duties (Meffe, 2001). Practitioners are often juggling multiple conservation projects with limited funding, short-term contracts, little or no time allocated to writing-up and publishing results, and limited incentives as other conservation priorities sit higher up on their agenda (O’Donnell, Supp & Cobbold, 2010). Furthermore, whilst there are growing calls for greater publication and prioritisation of conservation studies focusing on solutions and interventions (Christie et al., 2020, 2021), the conservation science community has generally struggled to prioritise the generation of this type of scientific evidence (Williams, Balmford & Wilcove, 2020), which could be a more fundamental reason as to why this discipline suffers from longer publication delays than many others.

At the same time, writing-up and publishing studies of interventions is not easy. Even after write-up, a manuscript may be rejected from several journals or sources, including for subjective reasons regarding the perceived level of reader interest rather than the strength of results or their importance for conservation. Substantial edits are then required to suit different formats, referencing systems, and styles, whilst reviewers may suggest major changes which take time and resources to implement. Indeed, a survey of 60 ecological journals showed journal rejection rates averaged 54% (range of 23–82%; 95% Confidence Intervals = [50%, 58%]; extracted using WebPlotDigitizer (2020)), indicating that many studies will go through multiple submission processes (Aarssen et al., 2008). Such issues may be particularly difficult for individuals not publishing in their first language, where rejections and subsequent reviews can be more challenging and time-consuming (Pettorelli et al., 2021).

Our findings also suggest that write-up delay makes up a substantial proportion of overall publication delay, given that studies published in the non-peer-reviewed literature suffered from a total publication delay of 1.90 years, on average. We would also argue that write-up delay is likely to be longer than 1.90 years, rather than shorter, on average for peer-reviewed studies, given the need in the peer-reviewed literature to conform to specific styles, formats, and referencing requirements. Overall, our findings tentatively suggest that write-up delay is likely to be the largest form of delay in the publication delay process, followed by resubmission delay (the remainder of submission delay) and destination journal delay (Fig. 6). We suggest future work could build on our results by directly quantifying and disentangling the components of publication delay, to help target action in areas that require more focus.

How can we reduce publication delay?

In Table 1, we present a set of possible solutions that could help to reduce write-up, resubmission, and destination journal delay. Whilst these solutions focus specifically on conservation science, we believe they are relevant to many different disciplines tackling publication delay. Table S14 also shows delays (time from submission to acceptance, and time from acceptance to publication) at different journals from 2015–2021, which may be relevant to conservationists wishing to decide where to rapidly publish tests of conservation interventions (albeit with appropriate caveats on how representative these estimated delays are—see Methods).

The COVID-19 pandemic has seen a far-reaching response from the scientific community to boost the rate at which scientific research is being conducted and published (including studies of healthcare interventions) through clear incentives to publish, rapid peer-review, and streamlined editorial processes (Horbach, 2020). Whilst this acceleration has led to challenges (e.g., potential declines in methodological quality and retracted papers that could misinform decision-makers; Horbach, 2020), we believe that the conservation community could learn from this effort to build a strong evidence base that is rapidly updated with the latest studies of conservation interventions to help address the biodiversity crisis. Nevertheless, there is concern over the unavoidable trade-off between speed and quality in the dissemination of scientific evidence. For example, pre-prints (unpublished manuscripts deposited on pre-print servers) may make studies instantly accessible to decision-makers, but without rigorous peer-review, a cornerstone of the scientific publication process, such articles may contain poor quality data and analyses, and make unsubstantiated claims that are not supported by data. We believe it is crucial to minimise publication delay at each stage of the process, but not at the cost of reduced scientific rigour which may lead to poor quality evidence and ineffective, inefficient, or even harmful action.

Comprehensive and timely access to scientific evidence is vital for effective evidence-based decision-making in any mission-driven discipline, but particularly in biodiversity conservation given the need to reverse dramatic biodiversity losses. Concerted action is required to streamline the rigorous testing and reporting of conservation interventions’ effectiveness to cover known gaps and biases in the evidence base (Christie et al., 2020, 2021). Our study clearly demonstrates the need for academics, practitioners, journals, organisations, and funders to work together as a scientific community to reduce publication delay as much as possible.

Supplemental Information

Supplemental Information 1 Z-ratios and p-values from post-hoc statistical tests of differences in publication delay between studies from different Conservation Evidence synopses (each covering a distinct conservation subject—e.g., ‘Bird Conservation’).

Darker red or blue coloured cells indicate greater Z-ratios (blue = positive difference, red = negative difference) and larger diamonds indicate smaller p-values, whilst red coloured diamonds indicate p-values of p < 0.05 (black diamonds indicate p ≥ 0.05). For example, studies from the synopsis ‘Bee Conservation’ had a significantly shorter mean delay than studies from ‘Forest Conservation’ (top row, first column is dark red with a large red diamond), but did not have a significantly shorter mean delay compared to studies from ‘Sustainable Aquaculture’ (top row, second column from right is grey with a very small black diamond). Post-hoc tests of differences between synopses were conducted using Estimated Marginal Means with Tukey adjustment in the R package emmeans (Lenth, 2021, see main text).

Click here for additional data file.

Supplemental Information 2 Changes in publication delay relative to the year in which studies of interventions were published.

The shade of hexagons is relative to the number of data points (studies) at that position on the graph. The red solid and dotted lines represent Estimated Marginal Means (averaging over other explanatory variables) and associated 95% confidence intervals based on a quasi–Poisson Generalised Linear Model (GLM) for publication delay (see Table S8 for full model result). This supplemental figure presents all data used in the study (as opposed to Fig. 2, main text, which only visualises studies with a publication delay of 20 years or less). We conducted sensitivity analyses to check whether the trend changed in more recent decades (see Table S4).

Click here for additional data file.

Supplemental Information 3 Publication delay of studies of conservation interventions (in years) grouped by the IUCN Red List Category of the species that were studied.

Data presented for Amphibians (Amphibia from the Amphibian Conservation synopsis), Birds (Aves from the Bird Conservation synopsis), and Mammals (Mammalia from the Bat Conservation, Primate Conservation, and Terrestrial Mammal Conservation synopses) combined. IUCN threatened categories include Vulnerable, Endangered, and Critically Endangered, whilst non-threatened categories include Least Concern and Near Threatened (following IUCN Red List; 2020). We did not include the few studies on Data Deficient and Extinct in the Wild species (see Methods). Vertical solid lines show mean publication delay and dashed lines show 95% Confidence Intervals. Summary estimates were obtained using Estimated Marginal Means (averaging over other explanatory variables) based on quasi–Poisson Generalised Linear Models (GLMs).

Click here for additional data file.

Supplemental Information 4 Peer-reviewed and non-peer-reviewed publication sources analysed in the current paper containing studies in the Conservation Evidence database.

Click here for additional data file.

Supplemental Information 5 Mean publication delay (Estimated Marginal Means (EMMs)), associated 95% Confidence Intervals (CI), and numbers of studies for each Conservation Evidence synopsis (a collection of studies based on the conservation subject in which interventions have been).

EMMs values and 95% CIs are presented in Fig. 1 (Main Text) and were derived from a quasi–Poisson GLM (see Methods in Main Text) with three explanatory variables (synopsis, publication date, and peer-review category). The number of studies for each synopsis may not match the Conservation Evidence website as the database is being dynamically updated with more studies over time. In addition, studies can be present in multiple synopses, and for the purposes of our analyses we also excluded reviews and meta-analyses, as well as studies with no end date of data collection.

Click here for additional data file.

Supplemental Information 6 Selection of model structure by qAIC ranking.

Selected model was the full one with all three explanatory variables of peer-review, publication date, and synopsis. Publication delay (difference between year data collection ended and year study was published) was the response variable.

Click here for additional data file.

Supplemental Information 7 Results of sensitivity analysis using quasi–Poisson Generalised Linear Models (GLM) for different time periods to assess whether the fact that more recently published studies are more likely to have a longer delay may have affected our results (see Method).

The synopsis ‘Management of Captive Animals’ and publication source ‘non-journal literature’ were set to the zero value of the beta slope and used as reference categories. Values of 0.000 represent values less than 0.001.

Click here for additional data file.

Supplemental Information 8 Selection of model structure by qAIC ranking for taxonomic analyses.

Selected model for all taxa combined (amphibians, birds, and mammals) was the full one with all four explanatory variables of peer-review, publication date, synopsis, and IUCN Red List category. Selected models for amphibians and mammals included peer-review and publication date, whilst the selected model for birds only included IUCN Red List category. Publication delay (difference between year data collection ended and year study was published) was the response variable for all models. Models within <2 units of qAIC were decided between by selecting the most simple model structure (i.e., selecting the most parsimonious model).

Click here for additional data file.

Supplemental Information 9 Results of quasi–Poisson Generalised Linear Model (GLM) (see Methods) for taxonomic sensitivity analyses (selecting a random species from each study rather than the most threatened species).

The synopsis ‘Management of Captive Animals’ and publication source ‘non-peer-reviewed’ were set to the zero value of the beta slope and used as reference categories as these had the lowest mean publication delay values based on preliminary exploration of the data. p-values of 0.000 represent p < 0.001.

Click here for additional data file.

Supplemental Information 10 Selection of model structure by qAIC ranking for taxonomic sensitivity analyses (selecting a random species instead of the most threatened species from each study).

Selected models are identical to those selected for the main taxonomic analyses (see Table S5). Publication delay (difference between year data collection ended and year study was published) was the response variable for all models.

Click here for additional data file.

Supplemental Information 11 Results of quasi–Poisson Generalised Linear Model (GLM) to test for predictors of publication delay.

The synopsis ‘Bee Conservation’ and publication source ‘non-peer-reviewed’ were set to the zero value of the beta slope and used as reference categories (see Methods). p-values of 0.000 represent p < 0.001.

Click here for additional data file.

Supplemental Information 12 Results of pairwise comparisons of Estimated Marginal Means, derived from a quasi–Poisson Generalised Linear Model (see Methods) to test for statistically significant differences between the publication delay of studies in different synopses.

Significance level = 0.05. p-values of 0.000 represent p < 0.001. Comparisons were undertaken using the R package emmeans using the Tukey adjustment (Lenth, 2021, see main text).

Click here for additional data file.

Supplemental Information 13 Results of pairwise comparisons of Estimated Marginal Means, derived from a quasi–Poisson Generalised Linear Model (see Methods), to test for statistically significant differences between the publication delay of studies from different literatures.

Estimate is the log odds difference between categories. Significance level = 0.05. p-values of 0.000 represent p < 0.001. Comparisons were undertaken using the R package emmeans using the Tukey adjustment (Lenth, 2021, see main text).

Click here for additional data file.

Supplemental Information 14 Results of quasi–Poisson Generalised Linear Model (GLM) (see Methods) for the main taxonomic analyses.

p-values of 0.000 represent p < 0.001.

Click here for additional data file.

Supplemental Information 15 Results of pairwise comparisons of Estimated Marginal Means to test for statistically significant differences between the publication delay of studies on species with different IUCN Red List statuses.

Estimate is the log odds difference between categories. Significance level = 0.05. p-values of 0.000 represent p < 0.001. Comparisons were undertaken using the R package emmeans using the Tukey adjustment (Lenth, 2021, see main text) and Estimated Marginal Means were derived from a quasi–Poisson Generalised Linear Model (see Methods).

Click here for additional data file.

Supplemental Information 16 Sensitivity analysis results of pairwise comparisons of Estimated Marginal Means to test for statistically significant differences between the publication delay of studies on species with different IUCN Red List statuses.

Instead of selecting the most threatened species from each study, we selected a random species from each study. Estimate is the log odds difference between categories. Significance level = 0.05. p-values of 0.000 represent p < 0.001. Comparisons were undertaken using the R package emmeans using the Tukey adjustment (Lenth, 2021, see main text) and Estimated Marginal Means were derived from a quasi–Poisson Generalised Linear Model (see Methods).

Click here for additional data file.

Supplemental Information 17 Median and mean delays (in days) of studies published in different journals from 2015–2021 that may be relevant to conservation scientists.

These include journals that have previously published tests of conservation interventions (based on our study’s dataset), plus those who name includes the terms ‘wildlife’, ‘environment’, ‘biology’, ‘ecology’, ‘wildlife’, and ‘zoology’. These data were generated using code available here: http://doi.org/10.5281/zenodo.45516 (Himmelstein & Powell, 2016). Delay type = ‘Acceptance’ refers to number of days from submission to acceptance, delay type = ‘Publication’ refers to the number of days from acceptance to online publication. We caution, however, that this data is based on studies for which data on delays could be extracted and may not be representative of delays at each journal, particularly where data was only available for a very small number of studies (see ‘Number of studies’ column) (Himmelstein, 2016; Himmelstein & Powell, 2016).

Click here for additional data file.

We thank all those involved in collating the Conservation Evidence database for the extensive work synthesizing the evidence base used to inform this work. We thank David Williams for helpful feedback on an original version of the manuscript, and Ashley Simpkins for helpful advice.

Additional Information and Declarations

Competing Interests

Author Contributions

Data Availability

The authors declare that they have no competing interests.

Alec P. Christie conceived and designed the experiments, performed the experiments, analyzed the data, prepared figures and/or tables, authored or reviewed drafts of the paper, and approved the final draft.

Thomas B. White conceived and designed the experiments, performed the experiments, analyzed the data, authored or reviewed drafts of the paper, and approved the final draft.

Philip A. Martin conceived and designed the experiments, performed the experiments, authored or reviewed drafts of the paper, collated the scientific studies and created the database that this study analysed, and approved the final draft.

Silviu O. Petrovan performed the experiments, authored or reviewed drafts of the paper, collated the scientific studies and created the database that this study analysed, and approved the final draft.

Andrew J. Bladon conceived and designed the experiments, performed the experiments, authored or reviewed drafts of the paper, collated the scientific studies and created the database that this study analysed, and approved the final draft.

Andrew E. Bowkett performed the experiments, authored or reviewed drafts of the paper, collated the scientific studies and created the database that this study analysed, and approved the final draft.

Nick A. Littlewood performed the experiments, authored or reviewed drafts of the paper, collated the scientific studies and created the database that this study analysed, and approved the final draft.

Anne–Christine Mupepele conceived and designed the experiments, authored or reviewed drafts of the paper, and approved the final draft.

Ricardo Rocha performed the experiments, authored or reviewed drafts of the paper, collated the scientific studies and created the database that this study analysed, and approved the final draft.

Katherine A. Sainsbury performed the experiments, authored or reviewed drafts of the paper, collated the scientific studies and created the database that this study analysed, and approved the final draft.

Rebecca K. Smith performed the experiments, authored or reviewed drafts of the paper, collated the scientific studies and created the database that this study analysed, and approved the final draft.

Nigel G. Taylor conceived and designed the experiments, performed the experiments, authored or reviewed drafts of the paper, collated the scientific studies and created the database that this study analysed, and approved the final draft.

William J. Sutherland conceived and designed the experiments, authored or reviewed drafts of the paper, and approved the final draft.

The following information was supplied regarding data availability:

The data and code are available at: alecchristie888. (2021). alecchristie888/publication_delay_manuscript: Post-review code (1.1). Zenodo. https://doi.org/10.5281/zenodo.5207520.

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
