# Peer review of "Reducing publication delay to improve the efficiency and impact of conservation science"

_PeerJ, doi:10.7717/peerj.12245_

## Round 0.1 · original submission · Minor Revisions

Though both reviewers suggested "minor revisions" the first reviewer has some questions about the statistical methods and the second reviewer about whether the sample is representative of the full dataset. These are important questions, which I want you to pay particular attention to.

·

Basic reporting

This study explores publication delay as a very important issue for practitioners and scientists, which causes so much headache and frustration. The authors use a huge and good-quality database to explain how publication delay changes over years and is affected by targeted species groups and their threat status, publication types and subjects. The paper is well written, methodologically sound and using sufficient background to support the authors' results. The authors describe the structure of publication delay and propose a useful pathway to reduce it without compromising the quality of science.

Below I suggest some ideas and comments on how this nice study could be improved and enriched.

The language is good, just some minor changes are needed:

Lines 44-45 – delete (time from completing data collection to publication) as this explanation is already given above
Line 46 – change “focussed” to “focused”. Add “significantly” or “marginally” to “smaller delay” to make it more informative
Line 61 – change “decision making” to “decision-making” throughout the text
Lines 69-70 – the sentence sounds incomplete and should be re-worded
Line 154 – change “in “unrecognised journals” ” to “an “unrecognised journal””
Lines 171-172 – this sentence is unclear, please explain or re-word: Any publication source that explicitly stated that it was a scientific journal was categorised as an ‘unrecognised journal’.
Line 194 – delete “will”
Line 197 – do you mean January 2001 to December 2001 or January 2003 to December 2003, to keep them within one year?
Line 216 – the categories of “Management of captive animals” and “non-journal literature” were set to the zero value of the beta slope as redundant, not set as the intercept, because models produce their own intercepts by default
Line 231 – add the comma after “To do this”
Line 233 – add “the” after “within”
Line 314 – change to “(Fig.S2), there were inconsistent, but significant differences in publication delay between”
Lines 348 and 365 – change 3.5 to 3.6
Line 425 – change 2.7 to 3.7 for recognized journals
Line 448 – change “etc.” to “and other duties”

Experimental design

The study design is appropriate. I have just several questions:

I am not yet convinced how the authors identified unrecognized journals. I have checked their list in supplementary Table S2 and found it to contain such well-known and respected journals as Galemys and South African Journal of Wildlife Research. Some other readers would possibly find other reputable journals on this list.

Peer review is applicable not only to journals (which of course would make a majority), but also to books and chapters. How did the authors categorize them in this case?

Why did the authors use Poisson and quasi-Poisson GLM if the response variable (publication delay) is not a count statistic? Tukey test has been used as a post-hoc test for normal data, but as the authors used quasi-Poisson, the data were assumed to be non-normal. Also, Tukey test is a kind of parametric t-test, and I have seen t statistics also in supplementary files describing the results of quasi-Poisson analysis. Please, explain this.

The Methods part is lacking information on how the performance of GLMs was assessed: p values, 95% CI of beta around the zero, odds ratios? I think the AIC-based model ranking would also be useful to see which factors of synopses, publication types, years, etc. are best to explain publication delay.

Validity of the findings

The results are valid and well explained. Here I would like to make the following suggestions:

The Results section contains many references to supplementary files, yet the key numbers are not indicated and the reader would have to jump to supplementary files and back to follow the track. This is not convenient. I would suggest to add some numbers to Results to make reading smooth – e.g., in lines 297-301 to add publication delay estimates for the periods 1912-1980, 1980-1990, 1990-2000 and 2000-2020.

Line 263 and thereafter – please write the ranges of p values, not p<0.05, p<0.01 etc.

As to me, information about publication delay between different IUCN statuses is one of the most important messages of this study, but it is hidden mostly in Table S9 which is big and out-of-the-way. I would suggest to re-make Table S9 as a series of heat maps in the main text showing different colors for different intervals of Tukey test statistics. This also can be done for other Tukey tests in the paper. I am just advising to dig out important information from supplementary files and move it to the main text.

Also, it would be important to add more numbers in Abstract, such as publication delays +/- error for captive animals, non-journal vs. journal publications.

Original eps files of figures were not labeled on axes and therefore were not understandable. I found fully prepared figures only in the pdf file of the submission. Is this a journal regulation or there is some other reason?

Additional comments

The paper is very interesting, important and timely, and I thank the authors for it. Here I would like to express some more thoughts which the authors could consider adding to this study.

Delays can be caused by journals because they fail to find suitable reviewers or even handling editors, even by the journals having numerous editors in their editorial boards. From my experience, keeping a manuscript for about half a year at the stages of “editor assignment” or “under review” is not uncommon and possibly even becomes a routine now during the pandemic times. As reviewers are not paid, they have little motivation to respond and meet deadlines. This can be particularly relevant to publications on conservation effectiveness which require multidisciplinary knowledge and skills from reviewers (ecology, study designs, statistics) and may experience particular delays in publication time because suitable reviewers can be unavailable or unknown to editorial boards. Again from my experience, several times I was asked by journals to recommend my colleagues who could have a capacity to review the effectiveness papers. This is also an indicator that effectiveness is a new topic for conservation which should be widely promoted in science and practice.

Publication delay adds to a delay of waiting for funding between the time when a project is submitted and the time when it is approved (this also can be a research topic). This makes conservation practitioners spend much of their time for waiting of published knowledge and/or money.

Apart from reluctance and long time required to publish, a strong disincentive to publish is insufficient knowledge of English. Such authors perceive rejections as a personal tragedy and may refuse to submit later on. I know that Oryx helped the scientists from developing countries to write scientific papers, but I do not know if it works now. Such initiatives are essential.

Conservation effectiveness does not fit many journals and the re-submission delay can be particularly long. A rapidly growing number of new journals focused on practical conservation (Ecological Solutions and Evidence, Conservation Science and Practice, Frontiers in Conservation Science) may reduce publication delay. If the journal Conservation Evidence had an impact factor, it would also help stimulate submissions.

Reviewer 2 ·

Basic reporting

This is an insightful paper that is well written and clearly thought through, and I would like to congratulate the authors for that. I am also happy to see this issue receiving more research attention as it is a key issue for authors, editors, publishers and funders to consider. I have nonetheless a number of hopefully helpful concerns I would like to bring to the authors. I detail them below.

INTRODUCTION
This is a well written section but I wonder if it could benefit from being more concise. For example is the paragraph starting Ln 77 necessary? I believe removing it would not impact the key narrative.

Ln 91-93 This example based on the CR strikes me as needing clarification. Is the argument that many species will have three generations in just a couple of years? Given what we know of publication delay in conservation, and assuming we can agree that the 10 years time interval is for the most part not relevant given average delays are much smaller, it would be good to see this point more clearly articulated.

Experimental design

METHODS
Ln 181-182 Why was the sample tested such a small proportion of the total dataset? I understand the accuracy was high but I am left wondering to what extent the test sample represents the wider dataset.

Ln220-223 I was curious as to why the effects of these different variables were looked into separately and not together. Would that not have helped us understand their relative importance?

Ln 224 I think it would be good to see explained in more detailed why more recently published studies may be more likely to suffer from delays. Is this an argument around COVID-19?

Ln 241 Using only the most threatened species can potentially provide a skewed vision of the results. How often was the most common status also the highest in the IURCN Red List. I believe providing more details around this is analysis crucial to allow the reader to interpret the results.

My main outstanding question for the Methods section was why there was no exploration of the delays at the outlet level. This is an important piece of information especially as latter on the authors recommend conservationists to pick those journals that are fastest. However, I am not aware of where that information is available and so that would be an important contribution.

Validity of the findings

No Comment

Additional comments

DISCUSSION
Ln 358-360 – I see these comparisons with other fields as useful but I suggest a little more nuance. For example a lot of the conservation literature is social science. This means often longer word counts and so a longer writing, peer-reviewing and revising process. To what extent to we need to take these comparisons with a pinch of salt? The assumption that mission driven disciplines have comparable literatures may not always hold.

Ln 374-377 I am unsure about by the explanation around the shorter delays for captive animal management. Several of the explanations advanced seem to me to be factors that are not related to the key metric which if I understand correctly starts counting from the end of the data collection only. At the same time the hypothesis the authors advance regarding the specialist journals could be tested given the dataset at hand correct? I think it would be good to see this explored a little more or see the authors simply acknowledge that its unclear what drives this result, if that is the case.

Ln 410-412 Given that we are talking of more than 25% of a year I would not dismiss this issue so readily. I would simply recognize this as a potential shortcoming of the methodology.

Ln 415-416 I understand this criticism but at the same time its not always helpful to simply label a process has too slow. Do the authors have views as to what would be an acceptable time from end of data collection to publication? I ask this as I think it would be good to have targets to aim towards.

Ln 420 -421 This reads to me as a little speculative. Why would you argue that given past results? Furter detail would be important.

Ln 450-452 This is undoubtedly true but does it help explain the disparities regarding other fields? In academia for example I would argue that there are many similarities between conservation and other fields yet we seem to be doing worse.. why is that?

Ln 463 Given how wide the variation is, I think this point should be made using an average measure. Otherwise I am not sure the number does support the assertion that follows.

Ln 465-467. This is assuming they are written all in the same way which is not my experience. Documents written for non-peer-reviewed outlets are written in a different style and often take a shorter time to prepare as they to not have to for example follow even broad structure of the journals of the field. This requires acknowledging

Ln 493-494 Indeed there have been many retractions but I think the most important questions whether that is more than expected? That seems implied but the reference does not quite support that statement in my view..

TABLE 1
• Currently there is a lot of repetition across sections, as some measures are relevant to multiple stages. I wonder if it would be more effective to have the table organized by proposed measure with columns on the right where you would tick the different stages of delay that measure is relevant to?
• Is the lack of guidelines on scientific writing really a limitation? I know of many resources on that issue so would appreciate a little more detail here on what is meant more precisely
• One option not discussed here is the potential for a direct route where an article is either accepted of rejected with no option for corrections. This is done in several economics journals for example. See https://weai.org/view/EI-No-Revisions
• Some of these suggestions seem too conceptual. I believe most authors believe they calibrate their submissions in a reasonable fashion the issue of course is that Editors and reviewers disagree (!). How would this recommendation be put in action?
• Similar for the item on considering deadlines for peer-review. I am not sure what is being advocated for in concrete terms. I would suggest focusing only on actionable suggestions.
• A similar point comes across when thinking of selecting the fastest journals. How would authors do that given the data is not always available? (also note that Biological Conservation is repeated twice)

---

## Round 0.2 · accepted · Accept

Reviewer 1 has a few minor corrections. Please make these changes in the final version.

·

Basic reporting

I thank the authors for doing a tremendous work to improve their manuscript. Having carefully read the manuscript and checked its supplementary materials, I am glad to accept this paper for publication. There are just a few very minor technical errors that should be corrected:

Abstract: I suggest not to use ±2SD in Abstract or to replace it with 95% confidence intervals similar to Results. This is just for consistency and clarity as this error term is not provided in Results and supplementary materials.

Line 144: change “The aim is to cover” to “The aim was to cover”

Line 354: remove the dot after 2.84.

Lines 600-601: delete (note these are not endorsements, but suggestions) as it is already mentioned on lines 527-528

Experimental design

Strong and appropriate

Validity of the findings

Strongly valid and well presented

Additional comments

No comments

Reviewer 2 ·

Basic reporting

Many thanks for the meaningful revisions and detailed replies to my queries. I have happy with the current version.

Experimental design

No comment

Validity of the findings

No comment